# Recovering Lithium from the Cathode Active Material in Lithium-Ion Batteries via Thermal Decomposition

**Shunsuke Kuzuhara [1],\*, Mina Ota [1], Fuka Tsugita [1] and Ryo Kasuya [2]**

[1] National Institute of Technology, Sendai College, 48 Nodayama, Medeshima-Shiote, Natori, Miyagi 981–1239, Japan; a1812606@sendai-nct.jp (M.O.); s1400332@sendai-nct.jp (F.T.)

[2] National Institute of Advanced Industrial Science and Technology, Shimo-Shidami, Moriyama-ku, Nagoya, Aichi 463-8560, Japan; ryo-kasuya@aist.go.jp

\* Correspondence: kuzuhara@sendai-nct.ac.jp; Tel.: +81-22-381-0333

**Abstract:** In this study, calcination tests were performed on a mixed sample of lithium cobalt oxide and activated carbon at 300–1000 °C under an argon atmosphere. The tests were conducted to discover an effective method for recovering lithium and cobalt from the cathode active material used in lithium-ion batteries. Additionally, the effect of soluble fluorine on the purification of lithium carbonate was investigated by the addition of lithium fluoride to an aqueous lithium hydroxide solution and a $CO_2$ flow test was performed. The lithium recovery was ≥90% when the calcination occurred at temperatures of 500–600 °C. However, the percent recovery decreased at temperatures ≥700 °C. It was demonstrated that in order to increase the recovery while maintaining 99% purity of lithium carbonate in the recovered material, it was imperative to increase the temperature of the solution and to limit the F/Li ratio (mass%/mass%) in the solution to a value that did not exceed 0.05.

**Keywords:** recovery; carbonation; lithium carbonate; lithium ion battery

## 1. Introduction

Lithium-ion batteries (LIBs) have favorable characteristics such as high voltage, high capacity, and low self-discharge capabilities. Consequently, LIBs have extensive applications in personal computers, smartphones, and motor vehicles. Approximately 500 million LIBs are currently manufactured in Japan annually [1]. However, LIBs are only functional for a few hundred charge-discharge cycles [2,3] and deteriorate within 2–3 years. Therefore, the manufactured LIBs should be discarded after a few years [4]. This estimate applies to Japan and is similar to that in other countries. The maximum recycling rate for the spent LIBs is currently 1% [5], with 95% of the LIBs being disposed of in landfills [6]. The inappropriate disposal of spent LIBs can result in severe environmental damage. Therefore, there is a considerable demand for an appropriate method of disposal and resource recovery [7–9].

The components of LIBs include cathode-active materials, such as lithium cobalt oxide ($LiCoO_2$), lithium nickel oxide ($LiNiO_2$), or $LiNi_{1/3}Co_{1/3}Mn_{1/3}$, an aluminum cathode collector, a graphite anode active material, a copper anode collector, an aluminum or stainless steel casing, a polypropylene or polyethylene separator, a lithium hexafluorophosphate ($LiPF_6$) electrolyte solution, and a polyvinylidene difluoride binder. In Japan, the annual quantity of cobalt consumed in the manufacture of LIBs is approximately 10,000 t [10]. The uneven distribution of resources indicated the occurrence of price fluctuations and the uncertainty of the reliability of supply [11]. Numerous investigations on the resource recovery have focused on the cathode active material [12–15]. The proposed recovery methods include a wet process [16–18], a dry process [19–21], an electrochemical method [22–24], bioleaching [25–27], crushing and selection [28], and mechanochemical reactions [29]. At least 99%

of the cobalt and nickel can be recovered by the wet process. However, the procedure was complex, and exerted a high environmental burden because strong acids were used. On the other hand, the dry process enabled the simple, high-throughput treatment of the toxic electrolyte solution [30], binders, and separators, which existed as impurities in the valuable metals. Methods that poorly separated the calcined materials resulted in recovery loss.

Although lithium is an indispensable component of LIBs, the absence of economic feasibility hindered the implementation of a resource recovery process [31]. However, the widespread use of electric and hybrid vehicles has significantly increased the price of lithium carbonate ($Li_2CO_3$) to a cost of $17,000/t in 2018, more than three times the cost in 2010 [32,33]. Consequently, lithium recovery processes have been proposed in recent years. For example, the $Li_2CO_3$ has been previously studied by means of carbon reduction of the cathode active material and water leaching of the calcined materials [34]. However, contamination by the fluorine derived from lithium hexafluorophosphate and polyvinylidene difluoride may prevent the attaining a $Li_2CO_3$ purity of 99%, which is the minimum purity required for industrial use [31]. Methods have been proposed for the removal of binder components by prior heat treatment [35,36]. However, heating releases toxic gases, such as hydrogen fluoride, phosphorus pentafluoride, and phosphoryl fluoride ($PO_3F$), from the lithium hexafluorophosphate [30,37], and organofluorine compounds from polyvinylidene difluoride [38]. The appropriate treatment of these side streams is required.

With the increasing use of LIBs, it is essential for a recycling society to utilize a metal recovery strategy that includes simple, high-throughput processing that is safe and carries a low environmental burden. It is also important to resolve problems related to the treatment of halogens included in the recovered materials, by exploring processes at lower temperatures than those previously reported, in order to reduce energy consumption [34].

In this study, we conducted tests on the separation and recovery of lithium and cobalt compounds from a model cathode active material by the carbon reduction method and attempted to elucidate the reaction mechanism. We also investigated the effect of the fluorine derived from the electrolyte solution and binders upon $Li_2CO_3$ purification by carbonating an aqueous lithium hydroxide monohydrate solution in the presence of fluorinated substances.

## 2. Materials and Methods

### 2.1. Lithium Leaching

#### 2.1.1. Cathode Model Sample

A cathode model sample was prepared using the lithium cobalt oxide ($LiCoO_2$) powder (Toshima Manufacturing Co., Ltd., Saitama, Japan) as the cathode active material and activated carbon powder (FUJIFILM Wako Pure Chemical Corporation, Osaka, Japan) as the reducing agent. The expected reaction is one in which two solids, lithium oxide ($Li_2O$) and cobalt (II) oxide (CoO), are formed as shown in Equation (1a). On the basis of this equation, the mol/mol ratio of the cathode active material and reducing agent ($C/LiCoO_2$) was set at 2.5. The temperature dependence of the Gibbs free energy change ($\Delta_r G^\circ$) associated with the reaction at 1 atm pressure is described by Equation (1b) [2].

$$2LiCoO_2 \text{ (s)} + C \text{ (s)} \rightarrow Li_2O \text{ (s)} + 2CoO \text{ (s)} + CO \text{ (g)} \tag{1a}$$

$$\Delta_r G^\circ = 191.655 - 0.2033T, \text{ kJ/mol (T = 298–2000 K)} \tag{1b}$$

#### 2.1.2. Calcination Test

Figure 1 illustrates the calcination test apparatus. A powdered sample (1 g) was mixed in an agate mortar, spread homogeneously on an alumina boat covered with gold foil, and inserted into the center of a quartz tube positioned inside an electric furnace (KTF-035N, Koyo Thermo Systems Co., Ltd., Nara, Japan). Argon was passed through the tube at a flow rate of 100 mL/min. The temperature

was increased to 100 °C for 10 min; thereafter, the temperature was increased at a rate of 10 °C/min to 300–1000 °C and maintained for 60 min. After that, the contents of the reaction tube were cooled in an argon atmosphere to obtain the calcined sample.

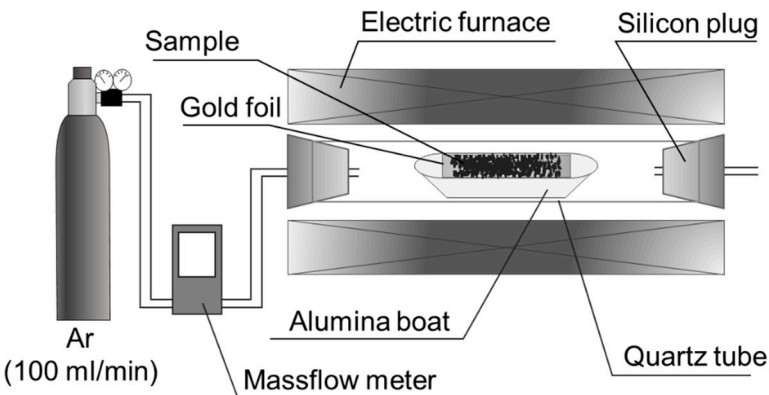

**Figure 1.** Overview of calcination test equipment.

### 2.1.3. Lithium Leaching Test

The calcined sample was crushed manually in an agate mortar and placed in a vial (internal diameter: 20.3 mm; external diameter: 40 mm; length: 120 mm). Ultrapure water (100 mL) was then added. The vial was placed in an ultrasonic cleaner and ultrasonicated for 1 h at room temperature. The dispersion liquid was filtered under reduced pressure using a polytetrafluoroethylene filter paper with a pore diameter of 0.1 μm. The filtered residue was dissolved in aqua regia, placed in a measuring cylinder and made up with 2.5% hydrochloric acid to the 250 mL mark. The filtrate was likewise made up to 250 mL.

### 2.2. Lithium Carbonate Purification

### 2.2.1. Dissolution Test

The following samples were prepared using lithium hydroxide monohydrate ($LiOH \cdot H_2O$) and lithium fluoride (LiF):

Lithium solution sample: $LiOH \cdot H_2O$ was dissolved in ultrapure water and the lithium concentration adjusted to 3000–10,000 ppm.

Fluorine solution sample: LiF was dissolved in ultrapure water and the fluoride concentration adjusted to 100–500 ppm. Subsequently, $LiOH \cdot H_2O$ was added and the lithium concentration of the solution adjusted to 5000 ppm.

### 2.2.2. Carbonation Test

The solution (300 mL) was transferred to a conical beaker with a stirring magnet and sealed with a silicone plug. The conical beaker was placed on a magnetic stirrer with heating capability, and the surrounding area covered with sea sand to conserve heat. A Teflon tube (6 mm in diameter) was inserted into the silicone plug until it came into contact with the solution. The temperature of the solution was measured using a thermocouple, and the heater output was adjusted to maintain the temperature within ±1 °C of the set value. When the temperature was set at 40 or 60 °C, the solution was sparged with carbon dioxide (250 mL/min for 60 min). Subsequently, the solution was filtered through the polytetrafluoroethylene filter paper, and both the filtrate and precipitate were recovered. The filtrate was made up to 500 mL with ultrapure water in a measuring flask to prepare the sample for analysis.

## 2.3. Characterization

### 2.3.1. Thermogravimetric (TG) and Differential Thermal Analysis (DTA)

The TG/DTA6300 apparatus (Hitachi High-Tech Science Corporation, Tokyo, Japan) was used to measure the cathode model samples. A 10 mg sample was placed in an alumina cell (2.5 mm in diameter) and the temperature was increased to 1000 °C at a rate of 10 °C/min under an argon flow of 300 mL/min. Aluminum oxide powder was used as the reference sample.

### 2.3.2. Crystalline Phase Identification

The crystalline phase of the sample was identified by powder X-ray diffraction (XRD) using a D8 ADVANCE/L (Bruker Co., Ltd., Karlsruhe, Germany). The measurement conditions were: 40 kV voltage; 40 mA current; $2\theta = 10$–$80°$; 0.02 step width; and 1 s/step counting time.

### 2.3.3. Lithium and Fluorine Quantification

The lithium content of the solution was measured by inductively coupled plasma atomic emission spectrometry (ICP-AES) using an ARCOS EOP (JEOL Co., Ltd., Tokyo, Japan). The fluoride content was measured using ion chromatography (DX-120 instrument from Dionex Co., Ltd., Tokyo, Japan).

### 2.3.4. Lithium Balance

The lithium solid/liquid/gas distributions after the calcination test were determined as follows:
Solid phase: The lithium in the filtered solid residue after the leaching test. It means unreacted solid residue and/or indissoluble in water.
Liquid phase: The lithium in the solvent after the leaching test. It means recoverable in water.
Gas phase: The gasification lithium. It was calculated by subtracting the lithium weight of solid and liquid phases from the total.

### 2.3.5. Carbon Analysis of Precipitate

The carbon content of the precipitate was measured using a carbon/sulfur analysis device (EMIA-920V, Horiba, Ltd., Kyoto, Japan). Tungsten (1.5 g) and tin (0.3 g) were added to the sample (10 mg) to improve the combustion. Complete combustion of the sample was achieved using oxygen as the carrier gas.

### 2.3.6. Calculation of Lithium Carbonate Concentration in the Precipitate

Mass balance calculations were done with the assumption that the precipitate formed by sparging with carbon dioxide was either $Li_2CO_3$ or LiF. The precipitate from the lithium solution was assumed to be $Li_2CO_3$. The three following steps were applied to the fluorine solution:

i.    The weight of F in the precipitate was calculated by subtracting the weight of F in the filtrate from the weight of added fluoride.
ii.   The weight of F in the precipitate was converted to the weight of LiF.
iii.  The $Li_2CO_3$ concentration in the precipitate was obtained by subtracting the weight of the LiF from the total weight of the precipitate.

The precipitation yield was defined as the weight ratio of precipitation/added reagent.

## 3. Results and Discussion

### 3.1. Thermal Behavior of Cathode Model Sample

The TG and DTA curves for the cathode model sample are shown in Figure 2. An exothermic reaction without weight loss occurred between 530 and 550 °C. An endothermic reaction with

rapid weight loss occurred between 660 and 780 °C. This behavior differed from that predicted by Equation (1a).

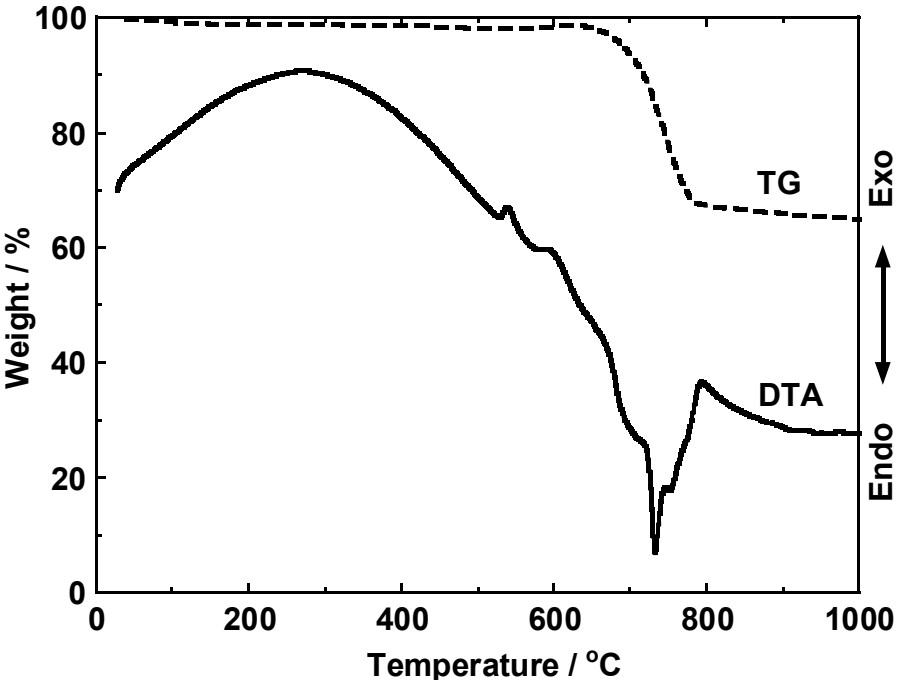

**Figure 2.** Thermogravimetric (TG) and Differential Thermal Analysis (DTA) curves for the cathode model sample.

Figure 3 shows the XRD profiles of the cathode model samples after calcination. The reaction at 400 °C was negligible, so only the peak for $LiCoO_2$ (PDF no. 70-2685) was observed. At 500 to 600 °C, peaks were observed for $Li_2CO_3$ (PDF no. 22-1141), Co (PDF no. 15-0806), and CoO (PDF no. 43-1004). At 700 °C and above, peaks were observed for $Li_2CO_3$, Co, and $Li_2O$ (PDF no. 12-0254).

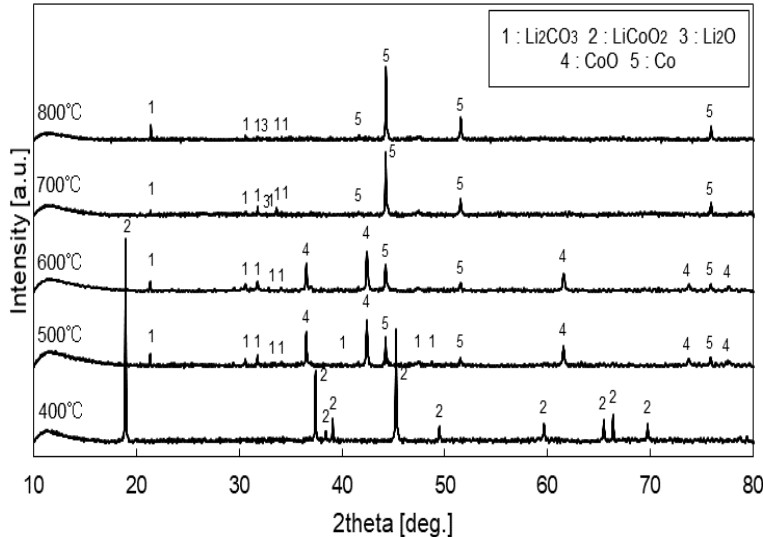

**Figure 3.** XRD profiles of cathode model samples after calcination.

### 3.2. Lithium Leaching Yield

Figure 4 depicts the variation of the lithium leaching yield and sample weight with the calcination temperature of the cathode model sample. At 300 and 400 °C, the maximum lithium leaching yield

was 2.6% and weight losses were 1.1% and 1.2%, respectively. At 500 and 600 °C, weight losses were 1.4% and 2.6%, respectively but the lithium leaching yield increased dramatically to 98.3% and 98.6%, respectively. At temperatures of 700 to 900 °C, the weight loss increased substantially, from 21.1% to 32.4%, whereas the lithium leaching yields ranged from 86.5% to 90.9%, which were lower than the yields at 500 and 600 °C.

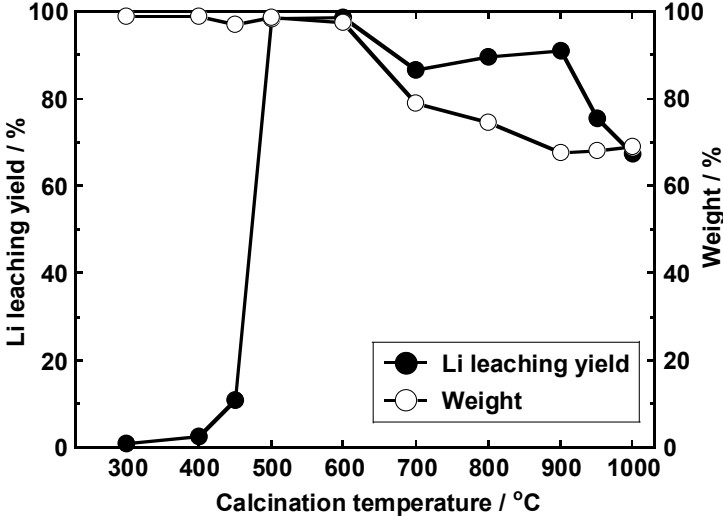

**Figure 4.** Variation of the lithium leaching yield and sample weight of the cathode model samples with the calcination temperature.

### 3.3. Lithium Balance

A comparison of the lithium gas/liquid/solid distributions of the cathode model sample at different calcination temperatures is presented in Figure 5. The fraction in the solid phase was 0.5–2.2% and exhibited minimal changes with respect to the calcination temperature. At 500 and 600 °C, the lithium was distributed solely between the solid and liquid phases. However, at temperatures of 700 °C and above, this two-phase partitioning was lost, as evidenced by the appearance of 8.5–11.3% lithium in the gas phase, which confirmed a relationship between the high-temperature calcination and the loss of lithium.

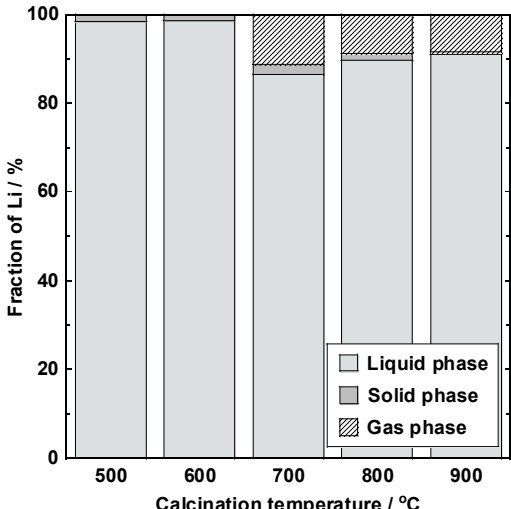

**Figure 5.** Lithium gas/liquid/solid distributions for the cathode model sample at different calcination temperatures.

### 3.4. Estimated Reaction Mechanisms

The proposed reaction mechanisms for various temperature ranges are described below.

### 3.4.1. At 520 to 550 °C

As indicated in Figures 2 and 3, the products of the $LiCoO_2$ calcination (irrespective of weight loss) are $Li_2CO_3$, Co, and CoO. Therefore, the presumed reaction in this temperature range is shown in Equation (2a) [2].

$$2LiCoO_2 \text{ (s)} + C \text{ (s)} \rightarrow Li_2CO_3 \text{ (s)} + Co \text{ (s)} + CoO \text{ (s)} \tag{2a}$$

$$\Delta_r G° = 65.805 - 0.08663T, \text{ kJ/mol (T = 298–2000 K)} \tag{2b}$$

To determine the reason for the preferential formation of $Li_2CO_3$ instead of $Li_2O$, the $\Delta_r G°$ values were calculated according to Equations (1b) and (2b) and compared. The temperature dependencies are shown in Figure 6. The $\Delta_r G°$ became negative for Equation (2b) at a temperature of 761 K (488 °C), which was significantly lower than that of Equation (1b) [943 K (670 °C)]. Therefore, calcination proceeded solely in accordance with Equation (2b) over this temperature range.

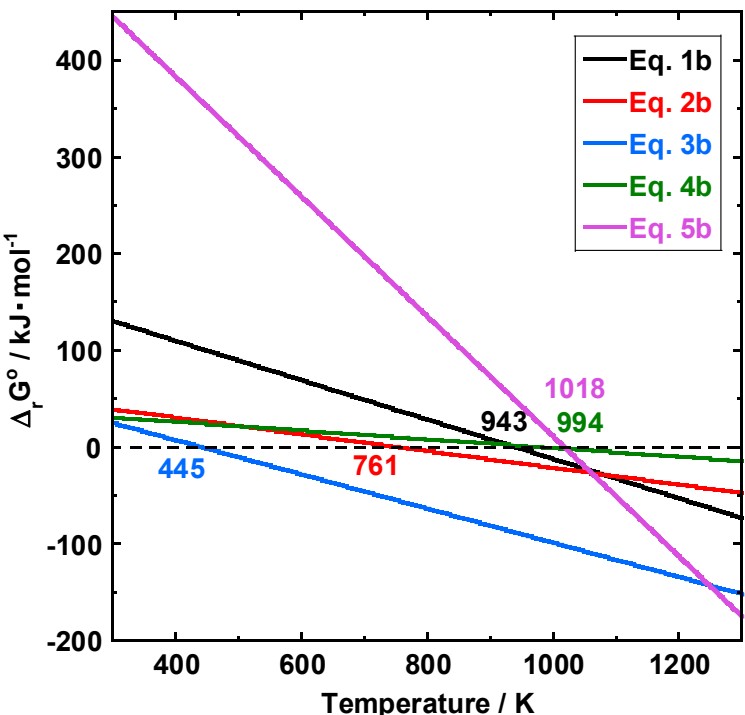

**Figure 6.** Temperature dependence of $\Delta_r G°$.

### 3.4.2. At 650 to 800 °C

The weight loss in this range was considered to be due to the reduction of CoO, as shown in Equation (3a) [2]. This reaction proceeded spontaneously because $\Delta_r G°$ became negative at 445 K (172 °C) (Figure 6). The weight loss calculated using Equation (3a) was 8.6%. A greater weight loss (30.8%) was found by the TG analysis (Figure 2). It is probable that there was a loss of lithium at 700 °C and above (Figure 5). The fusion of $Li_2CO_3$ (Equation (4a)) [39] was also proposed as a potential contributing factor. As expressed by Equation (4b) and Figure 6, the value of $\Delta_r G°$ for the melting of $Li_2CO_3$ became negative at 994 K (721 °C). In addition, $Li_2O$ formation (Equation (5a)) [2] may occur by the reduction of $LiCoO_2$ by carbon because the $\Delta_r G°$ for Equation (5b) was negative at 1018 K (745 °C).

$$2CoO(s) + C(s) \rightarrow 2Co(s) + CO_2 \text{ (g)} \tag{3a}$$

$$\Delta_r G^\circ = 78.52 - 0.17683T, \text{ kJ/mol (T = 298–2000 K)} \tag{3b}$$

$$Li_2CO_3 \text{ (s)} \rightarrow Li_2CO_3 \text{ (l)} \tag{4a}$$

$$\Delta_r G^\circ = 44.8 - 0.0451T, \text{ kJ/mol} \tag{4b}$$

$$4LiCoO_2 \text{ (s)} + 3C \text{ (s)} \rightarrow 2Li_2O \text{ (s)} + 4Co \text{ (s)} + 3CO_2 \text{ (g)} \tag{5a}$$

$$\Delta_r G^\circ = 631.07 - 0.61975T, \text{ kJ/mol (T = 298–2000 K)} \tag{5b}$$

### 3.5. Lithium Carbonate Recovery

The relationship between the lithium concentration and the $Li_2CO_3$ recovery is shown in Figure 7. $Li_2CO_3$ recovery was impossible at a lithium concentration of 3000 ppm regardless of the temperature of the solution. The aforementioned was observed because the amount of $Li_2CO_3$ formed was either at or below its aqueous solubility limit of 1.15 g/100 g at 40 °C and 0.99 g/100 g at 60 °C [40]. However, $Li_2CO_3$ recovery was possible when the lithium concentration was 5000 ppm. At solution temperatures of 40 and 60 °C, the $Li_2CO_3$ recoveries were 12% and 32%, respectively, which illustrated their temperature dependence. The maximum $Li_2CO_3$ recovery was 69% at 60 °C at a lithium concentration of 10,000 ppm.

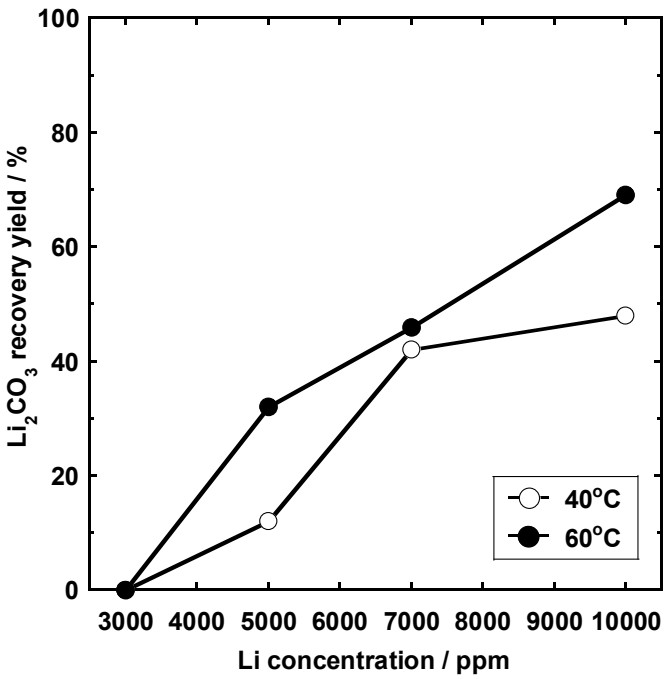

**Figure 7.** Relationship between $Li_2CO_3$ recovery and concentration of the lithium solution.

The XRD profiles of the $Li_2CO_3$ reagent and precipitate are shown in Figure 8a,b, respectively. The precipitate was obtained at a temperature of 60 °C and a lithium concentration of 5000 ppm. The precipitate peak was fully consistent with that expected for $Li_2CO_3$. The concentration of carbon in the precipitate determined via elemental analysis was 16.0 mass%, which equated to 98.5% when converted to $Li_2CO_3$ concentration.

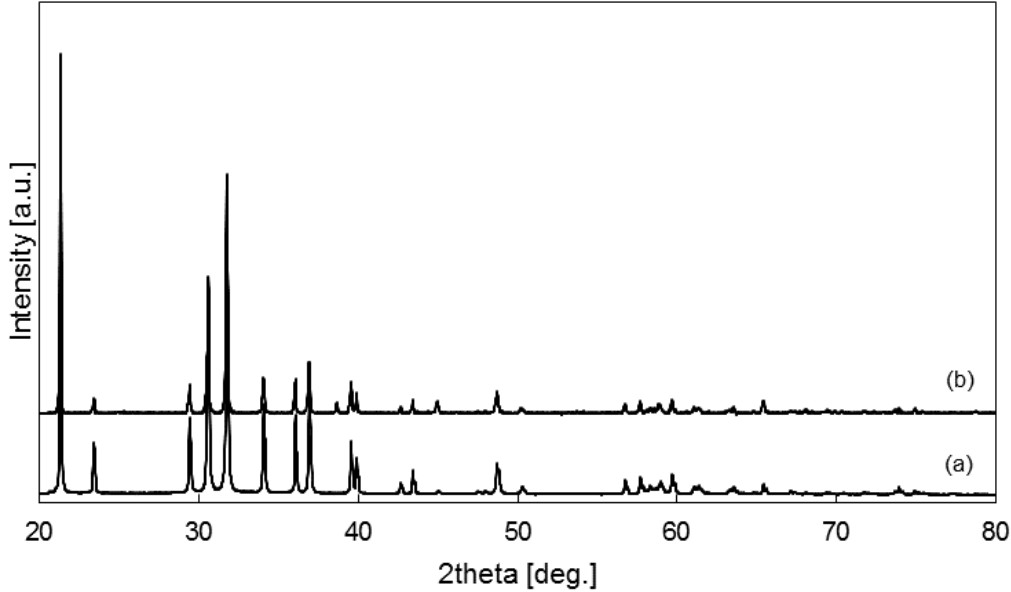

**Figure 8.** XRD profiles of (**a**) the $Li_2CO_3$ reagent and (**b**) the precipitate.

*3.6. Relationship between the Concentration of the Fluorine Solution and the Lithium Carbonate Concentration*

The relationship between the fluorine concentration in solution and the precipitation yield is shown in Figure 9. At 40 °C, the precipitation yield was low, and was independent of the concentration of the fluorine solution. At 60 °C, the precipitation yield increased with an increased concentration of the fluorine solution.

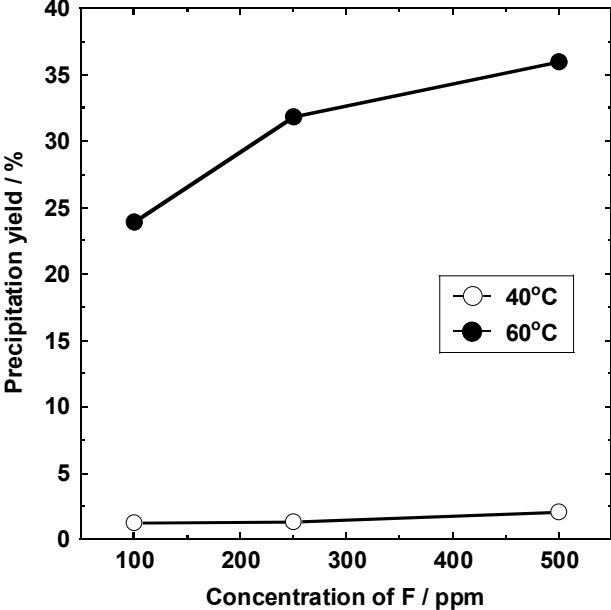

**Figure 9.** Relationships between solution fluorine concentration and precipitation yield.

The solid/liquid fluorine distributions at solution temperatures of 40 and 60 °C are shown in Figure 10. As the concentration of the fluorine solution increased, there was a tendency for the fluorine to be increasingly distributed in the solid phase. Fluorine was an impurity in the $Li_2CO_3$ purification process, which indicated that it was essential to maintain a relatively low concentration of the fluorine solution. However, at a fixed concentration of the fluorine solution, the temperature dependence of the solid/liquid fluorine distribution was negligible.

The Li$_2$CO$_3$ concentrations in the precipitates obtained from the fluorine solutions at temperatures of 40 and 60 °C are shown in Figure 11. It was necessary to maintain a relatively low concentration of the fluorine solution, regardless of the temperature of the solution, in order to achieve a high Li$_2$CO$_3$ concentration in the precipitate. It was also essential to increase the temperature of the solution to enhance recovery. It has been shown that to recover Li$_2$CO$_3$ efficiently at high purity (99%) by passing carbon dioxide, it was necessary to limit the F/Li ratio (mass%/mass%) in solution to a value that did not exceed 0.05. The F/Li ratio of discarded LIBs in the cathode materials was 0.68 [19]. Thus, considering that fluorine leaching also occurred during the lithium leaching process, it was necessary to either remove the fluorine beforehand or to convert it to a chemical form that did not undergo aqueous leaching.

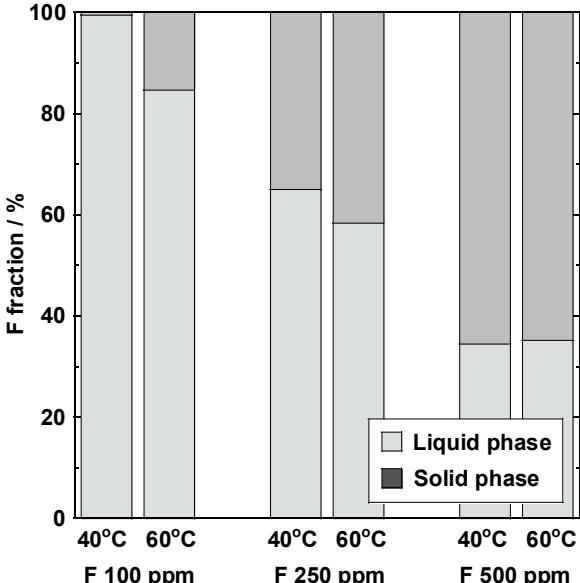

**Figure 10.** Solid/liquid fluorine distributions at temperatures of 40 and 60 °C.

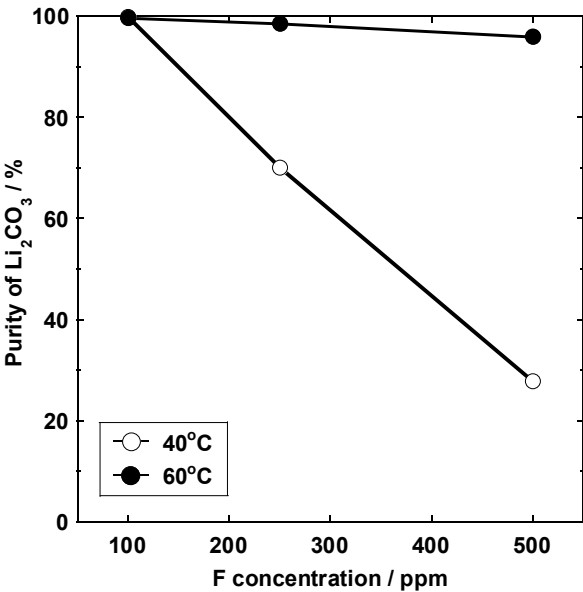

**Figure 11.** Li$_2$CO$_3$ concentrations in the precipitates obtained from fluorine solutions at 40 and 60 °C.

## 4. Conclusions

In this study, the separation and recovery of lithium and cobalt from the cathode active material used in LIBs was investigated by calcining a mixture of $LiCoO_2$ and activated carbon at 300–1000 °C, under an argon atmosphere. Additionally, LiF was added to an aqueous solution of lithium hydroxide and sparged with carbon dioxide to investigate the effects of the concentration of the fluorine solution on lithium purification. The minimum lithium recovery was 90% at 500–600 °C. $Li_2CO_3$ was present as a solid in this temperature range; therefore, the recovery yield was high. However, the formation of liquid $Li_2CO_3$ at 700 °C and above decreased the yield of lithium recovery. At 600 °C and below, the chemical form of the recovered cobalt was elemental Co or CoO, whereas at 700 °C and above, CoO was reduced by carbon, leaving only elemental Co as a residue.

Furthermore, we proved that in order to increase the recovery while maintaining 99% $Li_2CO_3$ purity, it was necessary to increase the temperature of the solution and to limit the F/Li ratio (mass%/mass%) in the solution to a value that did not exceed 0.05. It was also absolutely essential to introduce a method for excluding fluorine from the solution to achieve efficient, high-purity recovery of lithium from the discarded LIBs.

The influence concerning coexistence elements such as aluminum, copper and fluorine on the lithium recovery from spent LIBs should be investigated in the future. The lithium recovery process suggesting in this study is thought to be suitable for low value of cathode-active materials ($LiMn_2O_4$, $LiFePO_4$, etc.) from the viewpoint of its cost. However, the cost assessment for exhaust gas treatment during calcination and the optimization of lithium leaching are necessary for the industrial applications. Moreover the halogen recovery as resource should be considered in terms of cost reduction.

**Author Contributions:** Conceptualization, S.K. and R.K.; methodology, S.K. and R.K.; formal analysis, M.O. and F.T.; investigation, M.O. and F.T.; data curation, S.K. and R.K.; writing—original draft preparation, S.K. and R.K.; writing—review and editing, X.X. All authors have read and agreed to the published version of the manuscript.

**Funding:** This study was supported by the following grant: JSPS KAKENHI Grant Number 17H01925 and 19K12435. It was also funded by The Iwatani Naoji Foundation and Takahashi Industrial and Economic Research Foundation.

**Acknowledgments:** We would like to express our sincere gratitude to Takumi Akiyama, Masaya Hino, and Erika Furukawa (Tohoku University) who conducted the ICP-AES experiments.

**Conflicts of Interest:** The authors declare no conflict of interest.

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
