# Peer review of "Recovering Lithium from the Cathode Active Material in Lithium-Ion Batteries via Thermal Decomposition"

_metals, doi:10.3390/met10040433_

Round 1

Reviewer 1 Report

The manuscript presents "Heat treatment of the cathode active material of LIBs, Water leaching of Li, and Recovery of Li2CO3 by precipitation". The manuscript has been poorly organized and written. In addition, the subject of the heat treatment of the cathode active material followed by water leaching of Li is not new, resulting in the lacking of novelty. Some comments are given to improve the manuscript below.

  • - 3.1 Thermal Behavior of Cathode Model Sample should be explained quantitatively by combining with 3.4 Crystal Structures of Calcined Materials. Why are 3.1. and 3.4 separated? The reviewer suggests to combine 3.1. and 3.4 for the better interpretation of thermal Behavior of Cathode material
  • What is the residual content of F after heat treatment?
  • lithium gas/liquid/solid distributions of the cathode model sample ?? Explain in detail.
  • Please explain about water leaching of heat-treated cathode material in detail; Effect of water temperature, pulp density (ratio of water and cathode material) - Water leaching behavior of heat-treated cathode material should be explained with the solubility of Li2CO3
  • 3.3 Lithium Balance - The reviewer can not understand
  • In 3.6 Lithium Carbonate Recovery, what is the optimum conditions of water leaching of Li from heat-treated cathode material? And the maximum concentration of Li that is obtained under the optimum leaching conditions?
  • What is the reason to add the F?
  • Li2CO3 recovery should be interpreted with the solubility; effect of T and F ion
  • In Fig. 9, Why is the precipitation increased with F

Reviewer 2 Report

This paper deals with the investigation of lithium recovering from the cathode active material in lithium-ion batteries via thermal decomposotions. Authors reports a lithium recovery methodology using a model cathode active material and they also study the effects of the fluorine derived from the electrolyte solution and binders. While some of the results seem interesting and may warrant publication in Metals, current version of the manuscript comprises a number of flaws, which should be corrected before publication.

  1. Authors propose a methodology for the lithium recovery from a model cathode active material by thermal reduction using activated carbon powder. In addition to the active material and carbon black, a real cathode contains also a binder (PVFF) and contaminants from the electrolyte/separator. Which is the impact of these contaminants on the lithium recovery by the proposed approach by thermal reduction? In the introduction is explained that these contaminants release toxic gases after heating. Although authors study the impact of the fluorine in the electrolyte, they do not provide a methodology to remove the binder from the cathode material prior to the heat treatment. Further details and research in this point are welcome.
  2. Figure 4. How is the lithium gas/liquid/solid distribution obtained? The lithium in the solid is that obtained from the filtered residue after the lithium leaching tests? Was it measured by ICP? The lithium in the solvent is that in the filtrate after the lithium leaching test? How is it calculated the gas phase? Please, describe better in the manuscript the lithium balance for the cathode model sample at different calcination temperatures.
  3. Regarding the calculation of lithium carbonate concentration in the precipitate, is it possible to characterize/quantify the LiF ratio by XRD? In Figure 8 a small peak can be observed at around 37-38Ëš in the precipitate. Could it be related to LiF?
  4. Figure 9. The precipitation yield is the amount of F in the precipitate? Please, specify.
  5. The manuscript should be proofread since it contains several typos and the grammar should be checked.

Reviewer 3 Report

Comments to the authors:

  • All scientific papers should be written consistently in past tense, as the experiments were carried out in the past and not during reading of the manuscript/paper.
  • On p. 2, row 60, of the manuscript the authors use a phrase ‘byproduct’ which is not quite what they are looking for. A better expression would be ‘side stream’.
  • On p. 2, the decomposition reaction of lithium-cobalt double oxide has been written with CO(g) as product from 298-2000 K. This is slightly misleading as carbon monoxide is not stable below about 600 °C but decomposes to CO2(g) and elemental carbon.
  • In Eq. (1b) the unit of temperature may be K. That should be clarified in the equation, as well, e.g. T/K. The same comment is valid for Eq. (2b) - (5b), as well.
  • On p. 2, row 90, the authors use an expression ‘the calcination was complete’. What does that mean and how the end-point ‘complete’ was detected?
  • On p. 5, chapter 3.2, the authors discuss about the leaching kinetics and leaching rate of lithium from the calcine. The paragraph, however, uses the phrase ‘rate’ to describe the leaching yield or recovery. This is confusing. The same problem is in Conclusions where (rows 278-279) 90 % recovery is mentioned and in the following sentence its (reaction) rate is claimed to be high.
  • On p. 6-7, figure 4 and Chapter 3.3, the distribution of lithium between liquid, solid and gas phase is given. How the results were obtained from the experimental data?
  • On p. 8, rows 215-216, the authors end up with evaluation of the driving forces of reactions (3)-(5) and say that ‘DG for Li2CO3 melting becomes negative’. Why such a complicated expression and an explanation without outcome?
  • On p. 9-10, the recovery of lithium carbonate, obviously in leaching step of the experiments, is discussed. It is not clear, why 3000 ppm lithium was systematically lost and how lithium recovery was ‘possible when lithium concentration (in the solution) was 5000 ppm’? Where did it disappear and what is the origin of a correlation between lithium concentration in the solution and its yield?
  • On p. 11, row 247, it is not clear from the context what ‘precipitation yield’ means in this chapter.
  • Conclusions chapter, row 280, relates the high lithium recovery kinetics to be a result of solid Li2CO3 in the calcine. Where this conclusion can be drawn from?

Round 2

Reviewer 2 Report

Authors reviewed the manuscrupt and followed the suggestions. However, the lithium balance should be better explained.  Why is it necessary to have a solid/liquid distribution? I think that if the solid obtained from calcination would have digested in acid, the total lithium after calcination should be in the digested solution. 

Figure 4. Please check the text in the X axis. Calcination temperature.
